# Prevalence, predisposing factors, and turnover intention related to low back pain among health workers in Accra, Ghana

**Philip Apraku Tawiah** [1,2]*, **Emmanuel Appiah-Brempong**[3], **Paul Okyere**[3], **Geoffrey Adu-Fosu**[4], **Mary Eyram Ashinyo**[5,6]

1 Department of Occupational and Environmental Health & Safety, School of Public Health, College of Health Sciences, Kwame Nkrumah University of Science and Technology, Kumasi, Ghana, 2 Department of Pharmacognosy and Herbal Medicine, School of Pharmacy, University of Health and Allied Sciences, Ho, Ghana, 3 Department of Health Promotion & Disability Studies, School of Public Health, College of Health Sciences, Kwame Nkrumah University of Science and Technology, Kumasi, Ghana, 4 Physiotherapy Unit, Diagnostic and Rehabilitation Directorate, Ho Teaching Hospital, Ho, Ghana, 5 Department of Quality Assurance–Institutional Care Division, Ghana Health Service Headquarters, Accra, Ghana, 6 Department of Maternal and Child Health, Gilling's School of Global Public Health, University of North Carolina, Chapel Hill, North Carolina, United States of America

* ptawiah@uhas.edu.gh

**Data Availability Statement:** All relevant data are within the manuscript and its Supporting Information files.

## Abstract

### Background

Globally, low back pain (LBP) is responsible for disability among 60.1 million people. Health workers face a higher likelihood of being exposed to LBP compared to employees in the construction and manufacturing sectors. Data on LBP among hospital workers in Ghana are however limited. This study examined the prevalence, predisposing factors and turnover intention related to LBP among health workers in the Greater Accra region of Ghana.

### Methods

A multi-centred facility-based cross-sectional study was conducted in the Greater Accra region from January 30 –May 31, 2023. A multi-stage sampling technique was adopted, and the study participants were selected through proportion-to-size simple random sampling. STATA 15 software was used for analysis. Logistic regression analysis was used to determine the factors associated with LBP at a $p < 0.05$.

### Results

A survey was conducted among 607 health workers in 10 public and private hospitals. The prevalence of LBP was 81.6% [95% CI: (78.2%-84.6%)]. Advanced age [AOR = 1.07 (1.00, 1.16)], working for more than 5 days in a week [AOR = 8.14 (2.65, 25.02)], working overtime [AOR = 2.00 (1.16, 3.46)], rarely involved in transferring patients [AOR = 3.22 (1.08, 9.60)], most of the time involved in transferring patients [AOR = 6.95 (2.07, 23.26)], awkward posture during work [(AOR = 2.36 (1.31, 4.25)], perceived understaffing [(AOR = 1.84 (95% CI = 1.04–3.27)], sleep duration ≥ 8 [AOR = 0.54 (0.31, 0.97)] and sitting intermittently at work

**Funding:** The author(s) received no specific funding for this work.

**Competing interests:** The authors have declared that no competing interests exist.

[AOR = 0.31 (0.12, 0.80)] were factors significantly associated with LBP. A substantial number, 123 (24.9%), occasionally had intention of leaving their jobs.

## Conclusion

The study revealed a high proportion of low back pain (LBP), and turnover intention attributed to LBP. Moreover, organizational and occupational factors were found to be significantly associated with LBP. These findings underscore the importance of targeted interventions aimed at reducing the burden of LBP within these specific areas.

## Introduction

Low back pain (LBP) is a highly prevalent and incapacitating condition that affects a significant portion of the population, and it is also a common cause of musculoskeletal disorders [1–3]. LBP remains an overlooked public health problem and responsible for disability among 60.1 million people in the world [2,4] Additionally, it accounts for a number of disability-adjusted life years that is more than those caused by road traffic injuries, communicable and non-communicable diseases [5]. According to the 2010 Global Burden of Disease estimates, LBP ranked among the top ten diseases and injuries with the highest global burden [5]. Furthermore, it is estimated that governments and individuals, especially those in developing countries, spend up to $87.6 billion on medical treatment related to LBP [6].

Worldwide, the prevalence of LBP among adults stands at 40.0%, whilst the incidence rate is about 38.0% on yearly basis [7]. On account of occupational, physical and emotional factors, health workers are at more risk of experiencing LBP than workers of construction and manufacturing industries [8]. Obviously, the nature of work activities carried out by health workers makes them prone to LBP. Moreover, nurses are more exposed compared to other health personnel [9–11]. The main activities include repeated treatment sessions and rehabilitation procedures for patients, manual lifting, handling and transferring of patients, and working in an extreme awkward posture [12]. These tasks are extremely physically demanding, requiring intense workload and effort. Furthermore, in Africa and other low- to middle-income countries, the lack of working aids makes these tasks even more exhausting [13,14]. Some studies have predicted long-term sickness absence as one of the key consequences of LBP [15–17], which may heighten turnover intention among these professionals.

In a recent systematic review and meta-analysis work that included 154 studies, the pooled global lifetime prevalence of LBP among health workers was found to be 54.8% with gender, body mass index, physical exercise, and work-related variables as significant predisposing factors [18]. In another review conducted among nurses working in African clinical settings, prevalence rate of LBP was estimated at 64.1% with the lowest and highest prevalence found to be 44.1% and 82.7%, respectively [19]. Additionally, both lowest and highest prevalence rates were reported from a study conducted in Nigeria, a West African country. With respect to regions, West African region recorded the highest prevalence rate of 68.5% compared to the North and South African regions [19]. These estimates indicate the prevalence of LBP among health workers in West Africa, which includes countries like Ghana. Some studies in Ghana have highlighted the prevalence of turnover intention among health workers, with factors such as inadequate staffing, high workload, and burnout contributing to this phenomenon [20–22]. Additionally, occupational health and safety play a role in employee turnover intention, as job-

related stressors like low back pain can impact organizational commitment and job satisfaction, ultimately influencing turnover intentions [23].

Unfortunately, the menace of LBP is understudied in Ghana as evident in the scarcity of studies conducted within the past decade [24]. A recently published scoping review revealed that only two studies had tackled the issue of LBP among health workers in Ghana; and the prevalence rates were 49.5% and 51.2% [25,26]. These two unpublished studies in the review however estimated only the prevalence of LBP without considering the associated predisposing factors. In other words, there is no baseline study that assesses the prevalence and predisposing factors of LBP among health workers in Ghana. This study, therefore, investigated the prevalence, predisposing factors and turnover intention related to LBP among health workers in the Greater Accra region of Ghana.

## Materials and methods

### Study design, participants and setting

This study employed a facility-based analytic cross-sectional research design and a quantitative methodology. The study focused on doctors, nurses, midwives, laboratory staff, physiotherapists, healthcare assistant, orderlies and laundry staff across six public and four private hospitals located in the Greater Accra region of Ghana. These hospitals are major healthcare facilities within their respective districts, providing a range of services including outpatient departments (OPD) services, antenatal and family planning, dental care, eye care, laboratory services, ear-nose-and-throat care, radiology, dermatology services, and surgical procedures. The bed capacity of the hospitals varied from 50 to 500, and the total number of clinical and support staff ranged from 77 to 579. In 2015. The Greater Accra region was home to approximately 30.6% of all healthcare providers, including medical officers, midwives, nurses, and pharmacists, making it the region with the highest density of health workers [27]. In 2021, the Greater Accra Region was also the most populous region in Ghana, with an estimated population of 5,455,692, accounting for approximately 17.7% of the country's total population [28].

### Sample size determination

The Cochran formulae [29], $N_o = \frac{z^2 pq}{d^2}$, was used to determine the sample size for the study. Using z = constant for 95% confidence interval (CI) given as 1.96, p = proportion of the population (39.6%) that were exposed to LBP in a recent study conducted among health workers in Uganda [30], q = (1-p) and d = margin of error estimated as 5%, sample size, $N_o$ was estimated to be 368. After utilizing a design effect of 2.0 as recommended by previous similar studies [31,32], finite correction population formula proposed by Neyman [33,34] and an anticipated 10% non-response rate to the sample size, we arrived at final sample size of 652. However, 607 health workers participated in the study, resulting in a response rate of 93.1%. The main reason for not achieving 100% response rate was lack of monetary compensation.

### Sampling procedure

This study adopted a multi-stage sampling method. The Greater Accra region in Ghana was purposefully chosen, followed by random selection of districts, hospitals, and study participants. The selection of districts, hospitals, and participants was guided by a probability proportional-to-size sampling method. The Greater Accra region comprises 29 districts, including 2 metropolitan areas, 23 municipalities, and 4 districts. For this study, 10 districts, representing over 30.0% of the total, were selected. The sampling frame included 17 major hospitals, of which 10 were randomly chosen for the study. Each district was represented by one major

hospital, except in cases districts had two or three major hospitals where one hospital was randomly selected. The selection of major hospitals was influenced by the 2021 annual outpatient department (OPD) attendance data from the District Health Information Management System (DHIMS) [35]. Participants were recruited through stratified random sampling based on their respective profession. The professional groups served as strata, and study participants were randomly selected from them.

## Inclusion and exclusion criteria

The study involved participants including doctors, nurses, midwives, laboratory staff, physiotherapists, healthcare assistants, orderlies and laundry staff who had been working at a hospital for at least one year. Excluded were other healthcare professionals like administrators, radiologists, dieticians, and health students.

## Study questionnaire and data collection

The study questionnaire was purposely designed for the entire study: however, some portions were adapted from National Institute for Occupational Safety and Health, US Centre for Disease Control and Prevention's Healthcare workers Safety and Health Survey questionnaire [36], and a previous study [30]. The questionnaire comprised of both closed-ended and open-ended questions. The questionnaire was structured into five sections namely: Section I: Respondent's socio-demographic and lifestyle characteristics; Section II: Occupational factors; Section III: Organizational, behavioural and intervention factors; Section IV: LBP and Section V: Turnover intention with 12, 11, 5, 2 and 2 questions, respectively. To assess the study questionnaire's validity and reliability, the questionnaire was pre-tested among 60 health workers of the Ho Teaching Hospital. Following the pre-testing was review of questions based on suggestions from study participants, occupational health and safety faculty members, and senior management members of the Ghana Health Service.

The self-administered paper questionnaire in English Language was shared among selected study participants in their various departments at the hospital after a brief interaction with them. Participants were admonished to complete the questionnaire as early as they can; however, participants were given up to the next day to complete the questionnaire. In instances where participants requested for assistance regarding their inability to complete the study questionnaire on their own, research assistants administered the questionnaire in a form of interview. The information on completed paper questionnaire were entered into an earlier developed electronic platform, Open Data Kit [37]. The duration, January 30 –May 31, 2023, was used for recruitment of participants, and data collection.

## Data management and analysis

The data were exported from Open Data Kit electronic platform [37] and imported into STATA SE version 15 (64-bit) statistical analysis software [38] for cleaning and analysis. Participants involved in the pre-testing were excluded from the analysis. Preliminary analysis like frequencies was conducted on all variables to confirm the presence or absence of missing values. Also, skewness and kurtosis tests were performed on quantitative variables to determine their suitability for parametric or non-parametric tests.

Descriptive statistics such as frequencies and percentages were used to summarize categorical variables, whilst median and interquartile range were used for continuous variables. The descriptive statistics of the independent variables (socio-demographic and lifestyle characteristics, occupational factors, organizational, behavioural, and intervention strategies) were presented in the form of tables whereas that of the dependent variable (prevalence of LBP in the

past year), and duration of experiencing LBP, and turnover intention after experiencing LBP were presented in pie chart, bar graph and table, respectively. The outcome variable, prevalence of LBP, was evaluated by respondents indicating whether they had experienced LBP within the past year, with 'Yes' denoting affirmative responses and 'No' indicating the absence of such experiences. Also, participants were queried regarding the duration of their low back pain experience. Additionally, turnover intention was assessed among study participants who experienced LBP.

A Chi-square, Fisher's exact and Mann-Whitney U tests were used to determine preliminary associations between prevalence of LBP and independent variables. Additionally, variables significant at a p-value less 0.05 on the above tests were considered and confirmed on the bivariate and multiple logistic regression model. Statistical parameters including crude odds ratio, adjusted odds ratio, 95% CI and p-value were calculated based on a two-sided test.

### Ethical consideration

The study protocol was approved by Committee on Human Research Publication and Ethics (CHRPE) of Kwame Nkrumah University of Science and Technology, Kumasi with an approval reference number, CHRPE/AP/807/22, and the Ghana Health Service Ethics Review Committee with an identity number GHS-ERC:012/03/23. Study participants were briefed on potential risk/benefit, privacy and confidentiality, data storage and usage, voluntary consent/withdrawal, compensation, declaration of conflict of interest and research funding. Finally, participants read and completed a written informed consent form before permitted to take part in the study.

## Results

### Socio-demographic and lifestyle characteristics of health workers

Table 1 presents a summary of health workers sampled from 10 major hospitals in the Greater Accra region of Ghana. Out of the 607 health workers that participated in the study, the majority, 543 (89.3%) and 332 (54.7%) belonged to clinical staff category and nursing profession group, respectively. A little over half of the participants, 312 (51.4%) were within the 30–40-year-old bracket, and the median age was 32 years, with an interquartile range of 28–37 years. The dominant group of the participants, 499 (82.2%) were females, and almost half (49.4%) were married. Most of the study participants, 558 (91.9) had attained tertiary education. A greater portion, 283 (46.6%) had less than 5 years of working experience, and the median working experience was 5 years, with an interquartile range of 3–12 years. Also, 532 (87.6%) worked with public health facilities, and 512 (84.4) were permanent staff. Additionally, a greater number of participants, 493 (81.2) worked for 5 days and below within a week. Besides, more than one-tenth, 100 (16.5) were supervisors. Regarding lifestyle, 308 (50.7%) frequently exercised, and 436 (71.8%) slept for less than 8 hours daily.

### Occupational related factors contributing to low back pain

The majority, 310 (51.1%) of study participants worked for overtime. Also, about half (49.9%) were in a mix of day, evening and night shifts, followed by only day shift (46.5%). A substantial number of participants, 232 (38.2%) were assigned on call duties. Also, a dominant number, 570 (93.9%) were full time workers while approximately one-tenth 62 (10.2%) worked in multiple facilities. A greater number of participants, 322 (53.1%) occasionally experienced pressure from work. Additionally, a little over one-third 227 (37.4%) of them sometimes sit for long during work. Most participants, 266 (43.8%) were sometimes engaged in the lifting of heavy

**Table 1. Socio-demographic and lifestyle characteristics of health workers.**

| Characteristics | n | % |
|---|---|---|
| *Socio-demographic characteristics* | | |
| **Gender** | | |
| Female | 499 | 82.2 |
| Male | 108 | 17.8 |
| **Age** | | |
| Median (IQR) | 32.0 | 28.0–37.0 |
| Younger than 30 | 211 | 34.8 |
| 30–39 | 312 | 51.4 |
| 40–49 | 68 | 11.2 |
| 50 and older | 16 | 2.6 |
| **Professional category** | | |
| Doctor | 41 | 6.8 |
| Nurse | 332 | 54.7 |
| Midwife | 130 | 21.4 |
| Laboratory staff | 34 | 5.6 |
| Physiotherapist | 5 | 0.8 |
| Orderlies | 54 | 8.9 |
| Laundry staff | 2 | 0.3 |
| Healthcare Assistant | 9 | 1.5 |
| **Type of health worker** | | |
| Clinical staff | 543 | 89.3 |
| Supporting staff | 65 | 10.7 |
| **Marital status** | | |
| Single | 295 | 48.6 |
| Married | 300 | 49.4 |
| Divorced/Separated/widowed | 12 | 2.0 |
| **Highest educational level** | | |
| Primary/secondary | 49 | 8.1 |
| Tertiary | 558 | 91.9 |
| **Type of health facility** | | |
| Private | 75 | 12.4 |
| Public | 532 | 87.6 |
| **Working experience** | | |
| Median (IQR) | 5.0 | 3.0–12.0 |
| Less than 5 | 283 | 46.6 |
| 5–10 | 109 | 18.0 |
| Above 10 years | 215 | 35.4 |
| **Type of employment** | | |
| Contract | 95 | 16.0 |
| Permanent | 512 | 84.0 |
| **Current position** | | |
| No position | 473 | 77.9 |
| Supervisor | 100 | 16.5 |
| Head of Department | 34 | 5.6 |
| **Working days in a typical week** | | |
| 5 and below | 493 | 81.2 |
| Above 5 | 114 | 18.8 |

(*Continued*)

**Table 1.** (Continued)

| Characteristics | *n* | % |
|---|---|---|
| *Lifestyle characteristics* | | |
| **Exercise frequently** | | |
| No | 299 | 49.3 |
| Yes | 308 | 50.7 |
| **Daily hours of sleep** | | |
| Less than 8 | 436 | 71.8 |
| 8 and above | 171 | 28.2 |

IQR–Interquartile range.

objects during their line of work, and a greater number, 213 (35.1%) sometimes found themselves involved in transferring patients. A significant number of participants, 318 (52.4%) worked in awkward posture while 286 (47.1%) sometimes maintain good posture (Table 2).

## Organizational, behavioural and intervention related factors contributing to low back pain

Two-third of participants, 401 (66.1%), mentioned they are understaffed in their department. Also, many of them, 240 (39.5%) used work procedures most of the time. With respect to intervention factors, a good number of participants, 496 (81. %) received training on work machinery, and more than half of them, 311 (51.2%) had also been trained on transport aids. Additionally, most of them, 382 (62.9%) had been given training on good working posture (Table 3).

## Proportion of LBP among Health Workers

As depicted in Fig 1, a huge number of participants, 495 (81.6%) [95% CI: (78.2%-84.6%)] experienced LBP in the past year. The majority of these cases of LBP, 206 (41.6%) lasted for less than a month (acute LBP) while a substantial number, 175 (35.4%) persisted for more than 3 months (chronic LBP) (Fig 2).

## Socio-demographic and lifestyle characteristics influencing LBP

A significant association was found between age (t = - 3.57, p < 0.001), working experience (t = - 2.41, p = 0.015), type of employment ($\chi 2$ = 12.90, p < 0.001), number of working days in a typical week ($\chi 2$ = 20.83, p < 0.001), and prevalence of LBP (Table 4). Regarding lifestyle characteristics, frequency of exercise ($\chi 2$ = 4.53, p = 0.033) and daily hours of sleep ($\chi 2$ = 12.91, p < 0.001) were found to be significantly related to prevalence of LBP (Table 4).

## Occupational factors influencing LBP

As shown in Table 5, a significant association was revealed between overtime ($\chi 2$ = 16.15, p < 0.001), on call duties ($\chi 2$ = 4.46, p = 0.035), and prevalence of LBP. Additionally, prolong sitting ($\chi 2$ = 18.90, p = 0.001), awkward posture ($\chi 2$ = 38.66, p < 0.001), and maintaining good posture ($\chi 2$ = 21.95, p < 0.001) were found to be significantly related to prevalence of LBP. Also, a significant relationship was shown between lifting heavy objects ($\chi 2$ = 14.54, p = 0.006), transferring patients ($\chi 2$ = 29.23, p < 0.001), and prevalence of LBP. Again, pressure from work ($\chi 2$ = 29.23, p < 0.001) was found to be associated with prevalence of LBP.

**Table 2. Occupational related factors contributing to low back pain.**

| Characteristics | N | % |
|---|---|---|
| **Overtime** | | |
| No | 297 | 48.9 |
| Yes | 310 | 51.1 |
| **Type of shift** | | |
| Day only | 282 | 46.5 |
| Evening/swing only | 16 | 2.6 |
| Night only | 6 | 1.0 |
| A mix of day, evening and nights | 303 | 49.9 |
| **On call duties** | | |
| No | 375 | 61.8 |
| Yes | 232 | 38.2 |
| **Type of employment** | | |
| Full time | 570 | 93.9 |
| Part time | 37 | 6.1 |
| **Work in multiple facility** | | |
| No | 545 | 89.8 |
| Yes | 62 | 10.2 |
| **Pressure from work** | | |
| Not at all | 28 | 4.6 |
| Occasionally | 322 | 53.1 |
| Frequently | 257 | 42.3 |
| **Prolong sitting** | | |
| Never | 63 | 10.4 |
| Rarely | 132 | 21.8 |
| Sometimes | 227 | 37.4 |
| Most of the time | 141 | 23.2 |
| Always | 44 | 7.3 |
| **Lifting heavy objects** | | |
| Never | 47 | 7.7 |
| Rarely | 148 | 24.4 |
| Sometimes | 266 | 43.8 |
| Most of the time | 108 | 17.8 |
| Always | 38 | 6.3 |
| **Transferring patients** | | |
| Never | 42 | 6.9 |
| Rarely | 89 | 14.7 |
| Sometimes | 213 | 35.1 |
| Most of the time | 184 | 30.3 |
| Always | 79 | 13.0 |
| **Awkward posture** | | |
| No | 289 | 47.6 |
| Yes | 318 | 52.4 |
| **Maintain good posture** | | |
| Never | 28 | 4.6 |
| Rarely | 104 | 17.1 |
| Sometimes | 286 | 47.1 |
| Most of the time | 168 | 27.7 |
| Always | 21 | 3.5 |

**Table 3. Organizational, behavioural and intervention related factors contributing to low back pain.**

| Characteristics | N | % |
|---|---|---|
| *Organizational factors* | | |
| **Understaffed** | | |
| No | 206 | 33.9 |
| Yes | 401 | 66.1 |
| *Behavioural factors* | | |
| **Use of work procedures** | | |
| Never | 13 | 2.1 |
| Rarely | 44 | 7.3 |
| Sometimes | 95 | 15.7 |
| Most of the time | 240 | 39.5 |
| Always | 215 | 35.4 |
| *Interventional factors* | | |
| **Trained on work machinery** | | |
| No | 111 | 18.3 |
| Yes | 496 | 81.7 |
| **Trained on transport aids** | | |
| No | 296 | 48.8 |
| Yes | 311 | 51.2 |
| **Trained on good working posture** | | |
| No | 225 | 37.1 |
| Yes | 382 | 62.9 |

## Organizational, behavioural and intervention factors influencing LBP

With respect to organizational factors, a significant association was revealed between understaffed ($\chi2$ = 3.95, p = 0.047), and prevalence of LBP. Also, regarding intervention factors, trained on transport aids ($\chi2$ = 4.05, p = 0.044) was found to be significantly related to LBP (Table 6).

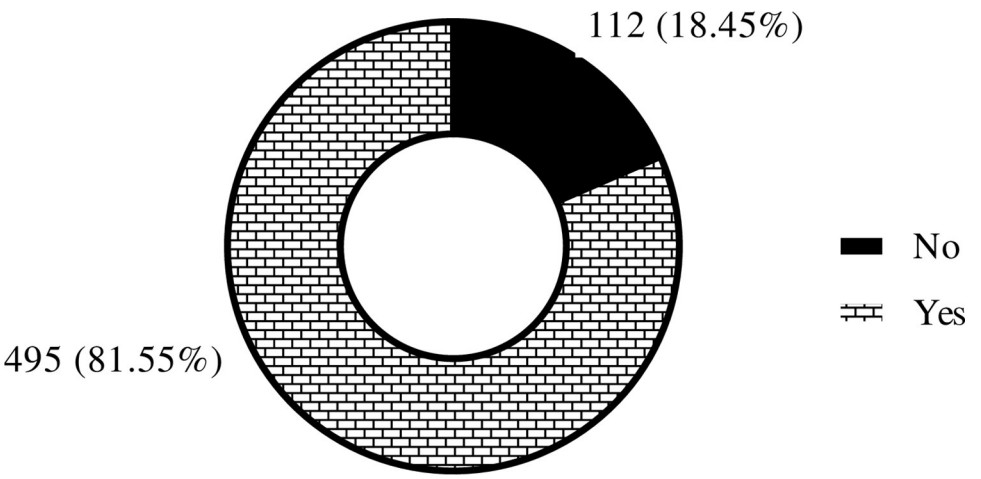

**Fig 1. Proportion of LBP among health workers.**

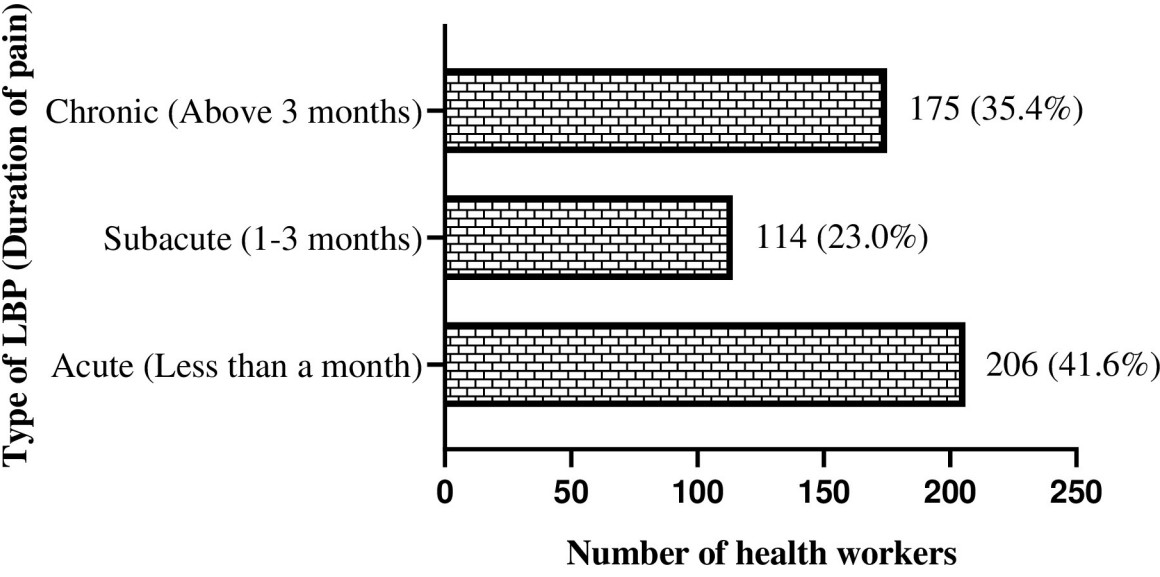

**Fig 2. Type of LBP experienced by health workers.**

## Bivariate and multiple logistic regression of predisposing factors and prevalence of LBP

The Table 7 below summarizes the bivariate and multiple logistic regression analysis between predisposing factors and prevalence of LBP. In the bivariate logistic regression, factors such as age, type of employment, frequency of exercise, daily hours of sleep, working days in a typical week, overtime, on call duties, pressure from work, lifting heavy objects, transferring patients, awkward posture, maintain good posture, understaff, and trained on transport aids were associated with LBP at $p < 0.05$. However, on the multivariate logistic regression analysis, only age, daily hours of sleep, working days in a typical week, overtime, on call duties, prolong sitting, transferring patients, awkward posture, and understaff were related to LBP.

For every 1-year increase in age, the odds of experiencing LBP increase by 7% (AOR = 1.07, 95% CI = 1.00–1.16, p = 0.044). Working for more than 5 days in a typical week increased the odds of experiencing LBP by approximately 8 times (AOR = 8.14, 95% CI = 2.65–25.02, p < 0.001) compared to health workers who worked fewer than 5 days. Additionally, working for overtime increased the odds of experiencing LBP by 2 times (AOR = 2.00, 95% CI = 1.16–3.46, p = 0.013) than individuals who worked for no overtime. Again, health workers who were most of the time, and rarely involved in transferring patients increased their odds of experiencing LBP by closely 7 times (AOR = 6.95, 95% CI = 2.07–23.26, p = 0.002) and 3 times (AOR = 3.22, 95% CI = 1.08–9.60, p = 0.035), respectively, compared to those never engaged in the transfer of patients. Also, working in an awkward posture increased the odds of experiencing LBP by approximately 2 times (AOR = 2.36, 95% CI = 1.31–4.25, p = 0.004) than those who did not work in an awkward posture. Further, workers who are understaff in their department increased their odds of experiencing LBP by almost 2 times (AOR = 1.84, 95% CI = 1.04–3.27, p = 0.036).

However, getting a daily sleep duration of 8 hours or more reduced the odds of experiencing LBP by 46% (AOR = 0.54, 95% CI = 0.31–0.97, p = 0.038) in comparison with individuals who slept less than 8 hours. And the ability to engage in periods of sitting for some time during working hours decreased the odds of experiencing LBP by approximately 69% (AOR = 0.31, 95% CI = 0.12–0.80, p = 0.015) compared to individuals who never sat for long periods.

**Table 4. Socio-demographic and lifestyle characteristics influencing LBP.**

| Characteristics | N | LBP | | χ2 | p |
|---|---|---|---|---|---|
| | | No | Yes | | |
| **Gender** | | | | 0.71 | 0.401 |
| Female | 499 | 89 (17.84) | 410 (82.16) | | |
| Male | 108 | 23 (21.30) | 85 (78.70) | | |
| **Age** | | | | - 3.57 | < 0.001*[b] |
| Median (IQR) | 32.0 (28.0–37.0) | 30.0 (26.0–35.0) | 32.0 (28.0–37.0) | | |
| **Type of health worker** | | | | 1.03 | 0.311 |
| Clinical staff | 543 | 103 (19.00) | 439 (81.00) | | |
| Supporting staff | 65 | 9 (13.85) | 56 (86.55) | | |
| **Marital status** | | | | 2.87 | 0.264[a] |
| Single | 295 | 57 (19.32) | 238 (80.68) | | |
| Married | 300 | 55 (18.33) | 245 (81.67) | | |
| Divorced/separated/widowed | 12 | 0 (0.00) | 12 (100.00) | | |
| **Highest educational level** | | | | 2.41 | 0.176 |
| Primary/secondary | 49 | 5 (10.20) | 44 (89.90) | | |
| Tertiary | 558 | 107 (19.45) | 451 (80.82) | | |
| **Type of health facility** | | | | 1.01 | 0.315 |
| Private | 75 | 17 (22.67) | 58 (77.33) | | |
| Public | 532 | 95 (17.86) | 437 (82.14) | | |
| **Working experience** | | | | - 2.41 | 0.015*[b] |
| Median (IQR) | 5.0 (3.0–12.0) | 4.0 (2.0–10.0) | 5.0 (3.0–12.0) | | |
| **Type of employment** | | | | 12.90 | < 0.001* |
| Contract | 95 | 30 (31.58) | 65 (68.42) | | |
| Permanent | 512 | 82 (16.02) | 430 (83.98) | | |
| **Current position** | | | | 2.30 | 0.317[a] |
| No position | 473 | 89 (18.82) | 384 (81.18) | | |
| Supervisor | 100 | 20 (20.00) | 80 (80.00) | | |
| Head of Department | 34 | 3 (8.82) | 31 (91.18) | | |
| **Exercise frequently** | | | | 4.53 | 0.033* |
| No | 299 | 45 (15.05) | 254 (84.95) | | |
| Yes | 308 | 67 (21.75) | 241 (78.25) | | |
| **Daily hours of sleep** | | | | 12.91 | < 0.001* |
| Less than 8 | 436 | 65 (14.91) | 371 (85.09) | | |
| 8 and above | 171 | 47 (27.49) | 124 (72.51) | | |
| **Working days in a week** | | | | 20.83 | < 0.001*[a] |
| 5 and below | 493 | 108 (21.91) | 385 (78.09) | | |
| Above 5 | 114 | 4 (3.51) | 110 (96.49) | | |

* Significant variable (p < 0.05)

[a] p-values calculated from Fishers' exact test.

[b] p-values calculated from Mann-Whitney U test; IQR–Interquartile range.

## Turnover intention related to the experience of LBP among health workers

The Table 8 depicts the turnover intention of study participants after their experience of LBP. Out of the 495 (81.6%) that experienced LBP, a substantial number, 123 (24.9%) and 144 (29.1%) occasionally consider leaving their job, and willing to accept another job at same compensation, respectively.

**Table 5. Occupational factors influencing LBP.**

| Characteristics | N | LBP | | χ2 | p |
|---|---|---|---|---|---|
| | | **No** | **Yes** | | |
| **Overtime** | | | | 16.15 | < 0.001* |
| No | 297 | 74 (24.92) | 223 (75.08) | | |
| Yes | 310 | 38 (12.26) | 272 (87.74) | | |
| **Type of shift** | | | | 5.17 | 0.169a |
| Day only | 282 | 54 (19.15) | 228 (80.85) | | |
| Evening/swing only | 16 | 0 (0.00) | 16 (100.00) | | |
| Night only | 6 | 0 (0.00) | 6 (100.00) | | |
| A mix of day, evening and nights | 303 | 58 (19.14) | 245 (80.86) | | |
| **On call duties** | | | | 4.46 | 0.035* |
| No | 375 | 79 (21.07) | 296 (78.93) | | |
| Yes | 232 | 33 (14.22) | 199 (85.55) | | |
| **Type of employment** | | | | 0.90 | 0.342 |
| Full time | 570 | 103 (18.07) | 467 (81.93) | | |
| Part time | 37 | 9 (24.32) | 28 (75.68) | | |
| **Work in multiple facility** | | | | 0.29 | 0.590 |
| No | 545 | 99 (18.17) | 446 (81.83) | | |
| Yes | 62 | 13 (20.97) | 49 (79.03) | | |
| **Pressure from work** | | | | 16.93 | < 0.001* |
| Not at all | 28 | 7 (25.00) | 21 (75.00) | | |
| Occasionally | 322 | 77 (23.91) | 245 (76.09 | | |
| Frequently | 257 | 28 (10.89) | 229 (89.11) | | |
| **Prolong sitting** | | | | 18.90 | 0.001*a |
| Never | 63 | 11 (17.46) | 52 (82.54) | | |
| Rarely | 132 | 13 (9.85) | 119 (90.15) | | |
| Sometimes | 227 | 60 (26.43) | 167 (73.57) | | |
| Most of the time | 141 | 24 (17.02) | 117 (82.98) | | |
| Always | 44 | 4 (9.09) | 40 (90.91) | | |
| **Lifting heavy objects** | | | | 14.54 | 0.006*a |
| Never | 47 | 16 (34.04) | 31 (65.96) | | |
| Rarely | 148 | 27 (18.24) | 121 (81.76) | | |
| Sometimes | 266 | 53 (19.92) | 213 (80.04) | | |
| Most of the time | 108 | 14 (12.96) | 94 (87.04) | | |
| Always | 38 | 2 (5.26) | 36 (94.74) | | |
| **Transferring patients** | | | | 29.23 | < 0.001* |
| Never | 42 | 16 (38.10) | 26 (61.90) | | |
| Rarely | 89 | 14 (15.73) | 75 (84.27) | | |
| Sometimes | 213 | 52 (24.41) | 161 (75.59) | | |
| Most of the time | 184 | 15 (8.15) | 169 (91.85) | | |
| Always | 79 | 15 (18.99) | 64 (81.01) | | |
| **Awkward posture** | | | | 38.66 | < 0.001* |
| No | 289 | 83 (28.72) | 206 (71.28) | | |
| Yes | 318 | 29 (9.12) | 289 (90.88) | | |
| **Maintain good posture** | | | | 21.95 | < 0.001*a |
| Never | 28 | 11 (39.29) | 17 (60.71) | | |
| Rarely | 104 | 7 (6.73) | 97 (93.27) | | |
| Sometimes | 286 | 50 (17.48) | 236 (82.52) | | |

(*Continued*)

**Table 5.** (Continued)

| Characteristics | N | LBP | | χ2 | p |
|---|---|---|---|---|---|
| | | **No** | **Yes** | | |
| Most of the time | 168 | 41 (24.40) | 127 (75.60) | | |
| Always | 21 | 3 (14.29) | 18 (85.71) | | |

* Significant variable (p < 0.05)

a p-values calculated from Fishers' exact test.

## Discussion

This study examined prevalence, predisposing factors and turnover intention related to LBP among health workers. More than three-quarters, 495 (81.9%), of healthcare workers experienced LBP in the past year. A notable proportion of lower back pain (LBP), 175 (35.4%), persisted for longer than three months and were classified as chronic. Advanced age, working more than five days per week, and overtime work were linked to higher odds of experiencing LBP. Additionally, involvement in patient transfers, working in awkward postures, and being understaffed were also associated with increased odds of experiencing LBP. Nevertheless, getting 8 hours or more of sleep per day and sitting intermittently during work were linked to reduced odds of experiencing LBP. Among the health workers who reported experiencing LBP, a significant number, 24.9%, occasionally contemplates leaving their current job, while 29.1% express willingness to accept another job with similar compensation.

**Table 6. Organizational, behavioural and intervention factors influencing LBP.**

| Characteristics | N | LBP | | χ2 | p |
|---|---|---|---|---|---|
| | | **No** | **Yes** | | |
| **Understaffed** | | | | 3.95 | 0.047* |
| No | 206 | 47 (22.82) | 159 (77.18) | | |
| Yes | 401 | 65 (16.21) | 336 (83.79) | | |
| **Use of work procedures** | | | | 7.34 | 0.111a |
| Never | 13 | 3 (23.08) | 10 (76.92) | | |
| Rarely | 44 | 4 (9.09) | 40 (90.91) | | |
| Sometimes | 95 | 25 (26.32) | 70 (73.68) | | |
| Most of the time | 240 | 45 (18.75) | 195 (81.25) | | |
| Always | 215 | 35 (16.28) | 180 (83.72) | | |
| **Trained on work machinery** | | | | 0.02 | 0.888 |
| No | 111 | 21 (18.92) | 90 (81.08) | | |
| Yes | 496 | 91 (18.35) | 405 (81.65) | | |
| **Trained on transport aids** | | | | 4.05 | 0.044* |
| No | 296 | 45 (15.20) | 251 (84.80) | | |
| Yes | 311 | 67 (21.54) | 244 (78.46) | | |
| **Trained on good working posture** | | | | 0.58 | 0.446 |
| No | 225 | 38 (16.89) | 187 (83.11) | | |
| Yes | 382 | 74 (19.37) | 308 (80.63) | | |

* Significant variable (p < 0.05)

a p-values calculated from Fishers' exact test.

**Table 7. Bivariate and multiple logistic regression of predisposing factors and prevalence of LBP.**

| Characteristics | Prevalence of LBP (n = 607) | | | | |
|---|---|---|---|---|---|
| | N | COR (95% CI) | p | AOR (95% CI) | p |
| **Age** | | | | | |
| Median (IQR) | 32.0 (28.0–37.0) | 1.05 (1.02–1.09) | 0.004* | 1.07 (1.00–1.16) | 0.044* |
| **Work experience** | | | | | |
| Median (IQR) | 5.0 (3.0–12.0) | 1.02 (0.99–1.06) | 0.127 | 0.95 (0.88–1.03) | 0.235 |
| **Type of employment** | | | | | |
| Contract | 95 | 1 | | 1 | |
| Permanent | 512 | 2.40 (1.48–3.96) | < 0.001* | 1.27 (0.61–2.62) | 0.522 |
| **Exercise frequently** | | | | | |
| No | 299 | 1 | | 1 | |
| Yes | 308 | 0.64 (0.42–0.97) | 0.034* | 0.81 (0.48–1.36) | 0.426 |
| **Daily hours of sleep** | | | | | |
| Less than 8 | 436 | 1 | | 1 | |
| 8 and above | 171 | 0.46 (0.30–0.71) | < 0.001* | 0.54 (0.31–0.97) | 0.038* |
| **Working days in a typical week** | | | | | |
| 5 and below | 493 | 1 | | 1 | |
| Above 5 | 114 | 7.70 (2.78–21.40) | < 0.001* | 8.14 (2.65–25.02) | < 0.001* |
| **Overtime** | | | | | |
| No | 297 | 1 | | 1 | |
| Yes | 310 | 2.37 (1.55–3.65) | < 0.001* | 2.00 (1.16–3.46) | 0.013* |
| **On call duties** | | | | | |
| No | 375 | 1 | | 1 | |
| Yes | 232 | 1.60 (1.03–2.51) | 0.036* | 0.97 (0.54–1.74) | 0.910 |
| **Pressure from work** | | | | | |
| Not at all | 28 | 1 | | 1 | |
| Occasionally | 322 | 1.06 (0.43–2.60) | 0.897 | 1.71 (0.58–5.08) | 0.332 |
| Frequently | 257 | 2.73 (1.06–6.99) | 0.037* | 1.89 (0.60–5.97) | 0.277 |
| **Prolong sitting** | | | | | |
| Never | 63 | 1 | | 1 | |
| Rarely | 132 | 1.93 (0.81–4.61) | 0.135 | 0.94 (0.33–2.71) | 0.914 |
| Sometimes | 227 | 0.59 (0.29–1.20)) | 0.146 | 0.31 (0.12–0.80) | 0.015* |
| Most of the time | 141 | 1.03 (0.47–2.26) | 0.939 | 0.80 (0.30–2.17) | 0.671 |
| Always | 44 | 2.12 (0.63–7.14) | 0.227 | 1.07 (0.26–4.51) | 0.919 |
| **Lifting heavy objects** | | | | | |
| Never | 47 | 1 | | 1 | |
| Rarely | 148 | 2.31 (1.11–4.82) | 0.025* | 1.54 (0.54–4.37) | 0.421 |
| Sometimes | 266 | 2.07 (1.05–4.07) | 0.034* | 0.90 (0.31–2.65) | 0.850 |
| Most of the time | 108 | 3.47 (1.52–7.90) | 0.003* | 0.69 (0.20–2.46) | 0.569 |
| Always | 38 | 9.29 (1.98–43.62) | 0.005* | 3.55 (0.54–23.23) | 0.186 |
| **Transferring patients** | | | | | |
| Never | 42 | 1 | | 1 | |
| Rarely | 89 | 3.30 (1.42–7.67) | 0.006* | 3.22 (1.08–9.60) | 0.035* |
| Sometimes | 213 | 1.91 (0.95–3.82) | 0.070 | 2.10 (0.76–5.82) | 0.154 |
| Most of the time | 184 | 6.93 (3.06–15.69) | < 0.001* | 6.95 (2.07–23.26) | 0.002* |
| Always | 79 | 2.63 (1.13–6.08) | 0.024* | 1.18 (0.36–3.92) | 0.781 |
| **Awkward posture** | | | | | |
| No | 289 | 1 | | 1 | |

(*Continued*)

**Table 7.** (Continued)

| Characteristics | Prevalence of LBP (n = 607) | | | | |
|---|---|---|---|---|---|
| | N | COR (95% CI) | p | AOR (95% CI) | p |
| Yes | 318 | 4.02 (2.53–6.35) | < 0.001* | 2.36 (1.31–4.25) | 0.004* |
| **Maintain good posture** | | | | | |
| Never | 28 | 1 | | 1 | |
| Rarely | 104 | 8.97 (3.04–26.37) | < 0.001* | 3.33 (0.87–15.21) | 0.078 |
| Sometimes | 286 | 3.05 (1.35–6.92) | 0.007* | 1.40 (0.48–4.13) | 0.541 |
| Most of the time | 168 | 2.00 (0.87–4.62) | 0.103 | 1.49 (0.50–4.46) | 0.475 |
| Always | 21 | 3.88 (0.92–16.36) | 0.065 | 2.93 (0.51–17.04) | 0.230 |
| **Understaffed** | | | | | |
| No | 206 | 1 | | 1 | |
| Yes | 401 | 1.53 (1.00–2.33) | 0.048* | 1.84 (1.04–3.27) | 0.036* |
| **Trained on transport aids** | | | | | |
| No | 296 | 1 | | 1 | |
| Yes | 311 | 0.65 (0.43–0.99) | 0.045* | 0.66 (0.39–1.11) | 0.115 |

* Significant variable (p-value < 0.05); IQR–interquartile range; COR–crude odds ratio; AOR–adjusted odds ratio.

In this present study, LBP prevalence of 81.6% was reported in the past year. This was parallel to studies conducted in Nigeria (87.3%) [39], South Africa (79.3) [40], Saudi Arabia (81.4%) [41] and China (80.0%) [42]. This similarity might be due to operational definition of LBP, study design, work settings and healthcare delivery system. Nonetheless, the proportion of LBP in this study was lower than a recent study conducted in Serbia (94.0%) [43] but higher than those conducted in Uganda (39.6%) [30] and Ethiopia (57.5%) [44]. This inconsistency may be due to socio-demographic characteristics of study participants, discrepancies in pain reporting culture and lifestyle. For instance, in Asian cultures, there is a cultural norm of anticipating a dramatic expression of emotion when confronted with pain, whereas in African societies, there is a preference for resilience, self-control, and minimizing the display of pain [45,46]. Furthermore, African individuals may perceive mild pain as a usual occurrence and might not readily report it as being in pain [47]. On the contrary, the proportion of LBP in this study was higher than a study carried out in Brazil (65.2%) [48], Uganda (39.6%) [30], Ethiopia

**Table 8. Turnover intention related to experience of LBP among health workers.**

| Variable | n | Percentage (%) |
|---|---|---|
| **Considered leaving your job** | | |
| Never | 168 | 33.9 |
| Rarely | 102 | 20.6 |
| Sometimes | 123 | 24.9 |
| Most of the times | 59 | 11.9 |
| Always | 43 | 8.7 |
| **Likely to accept another job at same compensation** | | |
| Never | 131 | 26.5 |
| Rarely | 107 | 21.6 |
| Sometimes | 144 | 29.1 |
| Most of the times | 62 | 12.5 |
| Always | 51 | 10.3 |

(57.5%) [49] and Nigeria (67.6%) [50]. Also, some studies conducted in Ghana recorded a prevalence of 73.3% [51], 51.2% [25] and 49.5% [26], which were lower than this present study. These variations may be due to different work settings, workload and the number of health facilities involved in a study. For instance, the previous studies in Ghana were conducted in one or two facilities in a region, and the workload of workers in those regions may be less and may have contributed to the low prevalence of LBP.

This current study found that an increase in age was associated with higher odds of experiencing LBP. This finding was similar to studies conducted in different parts of the world [6,12,18,44,52–55]. As health workers age, the risk of experiencing LBP increases due to degenerative changes in spinal discs, muscle weakness, reduced flexibility, slower recovery, and potential development of chronic conditions [56]. Also, as health workers advance in their careers, they may take on roles with higher levels of responsibility, which can include supervising and training junior staff, making critical decisions, and managing complex medical cases. These demanding roles may expose and put them at higher rate of experiencing LBP.

In this study, health workers who worked for 5 days and above had a higher odd of experiencing LBP compared to those who worked for less than 5 days. This outcome is in agreement with studies carried in Poland and the United States, where workers who worked for more than 40 hours a week were significantly at a higher risk of exposure to LBP [57,58]. This association could be attributed to factors like prolonged sitting or standing, repetitive movements, heavy lifting, inadequate rest breaks, poor ergonomics, and high levels of stress. Consequently, these factors can put strain on the lower back, leading to discomfort and pain [59]. Working more days without adequate breaks can cause persistent muscle fatigue, weakening the body's ability to recover from daily strain. Over time, this can lead to chronic lower back pain and discomfort due to decreased muscle strength and reduced joint mobility [60].

Additionally, health workers who worked overtime increased their odds of experiencing LBP in the current study. This finding is similar to a study conducted in Bangladesh and China [61,62]. Research shows that workers who work overtime or exceed eight hours a day are more likely to experience low back pain. A study revealed that these workers have almost double the risk of developing low back pain compared to those working standard hours [63]. The increased pain is linked to prolonged exposure to physical stressors, such as repetitive movements and poor body mechanics, commonly found in industrial environments. Overtime often involves prolonged manual handling tasks without adequate rest periods, resulting in repetitive strain. This strain leads to muscle fatigue and injury, especially in the lower back [64]. Prolonged exposure to heavy lifting and awkward postures over extended periods increases the risk of musculoskeletal disorders including low back pain. Health workers in most places may need to work overtime due to inadequate staff. This prolongs their already demanding work at the hospital and may lead to frequent development of LBP.

Besides, in this current investigation, workers in health facilities with inadequate staff were at a higher risk of experiencing LBP. The results of studies conducted by Kim et al. [65] and Sanjoy et al. [62] in the United States of America and Bangladesh, respectively, supported this finding. Also, insufficient clinical and supporting staff in a hospital could increase the number of manual handling tasks per worker, along with overtime work, consequently elevating the likelihood of experiencing LBP [62]. The issue of understaff is common to many healthcare settings, and it's not surprising that studies conducted in different settings had comparable results. Additionally, in a healthcare facility that is understaffed, individual staff members may have to handle a higher workload, including lifting and moving patients, without sufficient support or assistance [19]. This increased demand can lead to more frequent and prolonged manual handling tasks, which can put a strain on the lower back muscles and spine.

In this study, health workers who were involved in transferring patient had a greater odd of experiencing LBP than those who never transferred patient in their line of work. Similar studies carried out in Saudi Arabia [66] and Nigeria [12,50] have confirmed this finding. Health workers involved in transferring of patients often must lift, move, or reposition them, which can put immense strain on their lower back muscles and spine. Repeatedly performing these physical tasks, especially with improper body mechanics, can lead to musculoskeletal disorders, including LBP [67].

Again, according to the findings of this recent study, working in awkward posture increases the odds of experiencing LBP. This outcome of the research was comparable to the ones conducted among Ugandan and Ethiopian health workers [30,68]. These similarities in results may be due to similar ways of health workers in these African countries adopt unnatural or uncomfortable positions while sitting, standing, or lifting during work, which puts additional stress on the muscles, ligaments, and discs in the lower back. Moreover, prolonged periods of maintaining awkward postures can lead to muscle fatigue, strain, and increased pressure on the spine, increasing the likelihood of developing LBP.

Nonetheless, this current study also found that sleeping for a duration of 8 hours and above reduces the odds of experiencing LBP. This evidence is consistent with a study conducted in China, where it was found that longer sleeping hours decreased the experience of LBP [69,70]. The insufficient duration of sleep may affect all health workers in the globe and can significantly influences the chance of experiencing LBP. Insufficient or poor-quality sleep can hinder muscle recovery, increase pain sensitivity, impair cognitive function, and disrupt inflammation regulation [71]. Prioritizing adequate sleep and practicing good sleep hygiene can help reduce the risk of LBP and promote overall well-being.

In this study, health workers who had intermittent rest periods during work hours had reduced odds of experiencing LBP. This outcome is consistent with studies conducted in Turkey, Ethiopia and Norway [44,52,72]. Taking breaks or sitting intermittently during working hours help reduce muscle fatigue, improve circulation, maintain spinal alignment, decrease spinal compression, and promote movement, contributing to reduction of exposure to LBP, and better overall musculoskeletal health [73].

Though a couple of studies have predicted a strong association between LBP and long-term sickness absence [15–17,74], and motivation and job satisfaction on turnover intention [75], there is lack of studies that examined turnover intention after experiencing LBP among health workers. However, in this current study, a substantial number of study participants occasionally considered leaving their job and were willing to accept another job at same compensation, respectively, after experiencing LBP. Therefore, further studies are necessary to examine the effect of LBP on turnover intention in the midst of the recent rising number of health workers leaving the health industry, both in Ghana and the globe [75,76].

The findings of this study have significant implications for policy and practice in the healthcare sector. Policymakers and healthcare administrators should prioritize strategies to mitigate the risk of lower back pain (LBP) among healthcare workers, such as implementing ergonomic interventions, including enhanced ergonomic training and application through regular reminders, visual cues, and periodic "posture refreshers". Also, ensuring access to assistive lifting devices, such as mechanical lifts, and encouraging a culture that prioritizes employee health and safety by regularly addressing occupational health risks are also crucial. Additionally, establishing a reporting system for staff to communicate ergonomic concerns or report instances of musculoskeletal pain can facilitate prompt interventions. Further, policies limiting overtime work, ensuring adequate staffing levels, promoting healthy sleep habits, and encouraging regular breaks during work hours can help reduce the prevalence of LBP. Furthermore, healthcare institutions should also provide training on proper lifting techniques, patient

transfer protocols, and stress management to minimize the risk of LBP. By addressing these modifiable risk factors, healthcare organizations can reduce the burden of LBP, improve worker well-being, and maintain high-quality patient care.

## Strength and limitations

The research took place among health workers chosen from 10 major facilities in the National Capital of Ghana, aiming to mirror the overall situation in the country. However, there are certain constraints associated with this study. The utilization of a cross-sectional study approach means that it cannot establish definitive cause-and-effect relationships or ascertain the sequence of causation between different factors. Furthermore, the study is susceptible to recall bias, as participants were questioned about events from the previous 12 months. Also, the generalization of the study findings may be applicable to only major health facilities due to sampling of participants from only major hospitals.

## Conclusion

The prevalence of LBP among health workers in the Greater Accra region was high. Advanced age, working more than 5 days in a typical work week, working overtime, being involved in transferring patients, working in awkward posture, understaffed were significant associated with higher risk of exposure to LBP. On the contrary, sleeping for 8 hours or more, and ability to sit regularly during work were related to lower likelihood of experiencing LBP. A concerned proportion indicated a strong intention to leave their current job position due to experiences of LBP. Health administrators and policy makers should consider ways of addressing these occupational, organizational and demographic factors to reduce the exposure of health workers to LBP. Future research can utilize prospective cohort studies as research evidence in order to establish causal relationships, and further investigate the effect of LBP on turnover intention.

## Supporting information

**S1 File. The raw data supporting findings of the article.**
(PDF)

**S2 File. Data collection tool for the study.**
(XLSX)

## Acknowledgments

The authors extend their heartfelt gratitude to the study participants and management of the selected study health facilities who generously dedicated their time and resources to partake in this research study. Your invaluable cooperation and willingness to share insights have greatly contributed to the depth and significance of our findings.

## Author Contributions

**Conceptualization:** Philip Apraku Tawiah, Emmanuel Appiah-Brempong.

**Data curation:** Philip Apraku Tawiah.

**Formal analysis:** Philip Apraku Tawiah.

**Investigation:** Philip Apraku Tawiah.

**Methodology:** Philip Apraku Tawiah, Emmanuel Appiah-Brempong.

Project administration: Philip Apraku Tawiah.

Software: Philip Apraku Tawiah.

Supervision: Emmanuel Appiah-Brempong, Paul Okyere.

Visualization: Philip Apraku Tawiah.

Writing – original draft: Philip Apraku Tawiah.

Writing – review & editing: Emmanuel Appiah-Brempong, Paul Okyere, Geoffrey Adu-Fosu, Mary Eyram Ashinyo.

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
