## [Decision Letter · Decision Letter 0]

15 Mar 2024

PONE-D-23-26548Predisposing factors and turnover intention effects of low back pain among health workers in selected hospitals in the Greater Accra region, GhanaPLOS ONE

Dear Dr.Tawiah,

Thank you for submitting your manuscript to PLOS ONE. After careful consideration, we feel that it has merit but does not fully meet PLOS ONE’s publication criteria as it currently stands. Therefore, we invite you to submit a revised version of the manuscript that addresses the points raised during the review process.

We look forward to receiving your revised manuscript.

Kind regards,

Mohammad Hayatun Nabi, MBBS, MHSM, MPH, PHD

Academic Editor

PLOS ONE

Reviewers' comments:

Reviewer's Responses to Questions

**Comments to the Author**

1. Is the manuscript technically sound, and do the data support the conclusions?

Reviewer #1: Partly

Reviewer #2: Partly

2. Has the statistical analysis been performed appropriately and rigorously? 

Reviewer #1: Yes

Reviewer #2: Yes

3. Have the authors made all data underlying the findings in their manuscript fully available?

Reviewer #1: Yes

Reviewer #2: No

4. Is the manuscript presented in an intelligible fashion and written in standard English?

Reviewer #1: Yes

Reviewer #2: No

5. Review Comments to the Author

Reviewer #1: Please, see the attached report to see how you could revise the manuscript. Even as the statistical analysis was good, there appears to be some lack of clarity in respect of the focus or objective of the study. It was not very clear what the dependent variable of the study was.

Reviewer #2: Please note that in a cross-sectional design, causality cannot be established. Therefore, it would be more accurate to describe the observed relationship between turnover intention and low back pain as an association rather than an effect. Please, follow the same fashion throughout the manuscript.

Please clarify the abbreviation 'CI' for the readers. It stands for Confidence Interval. For instance, the reported prevalence of LBP was 81.6% [95% CI: (78.2% - 84.6%)]. Including the full form at least once at the beginning would enhance the clarity of your findings for the readers.

Please specify the unit for 'Sleep duration ≥ 8' to enhance clarity. For example, is it in hours? Providing this detail would help readers better understand the reported Adjusted Odds Ratio (AOR) of 0.54 (0.31, 0.97).

Consider specifying the type of interventions to be implemented in these areas to address the burden of LBP. Additionally, elaborating on the proposed mechanisms through which these interventions would operate and identifying the target population for these interventions would enhance the clarity and applicability of your recommendations.

Describing LBP as 'one of the most severe life-threatening burdens' of noncommunicable conditions may be overstated. While LBP is significant, it's important to maintain accuracy in descriptions. Noncommunicable diseases encompass a wide range of conditions, and LBP is not typically life-threatening like cardiovascular diseases or cancer.

In the introduction, it would be beneficial to adopt a consistent approach with abbreviations. After the first mention of each term, using the abbreviated form subsequently can streamline the text and improve readability. For example, after introducing 'low back pain (LBP)', subsequent references can use the abbreviation 'LBP'.

Consider refraining from using colloquial expressions like 'quite unfortunately' in academic writing. Instead, opt for more formal and precise language to maintain professionalism and clarity in your text.

Reporting and comparing different types of low back pain as simply 'low back pain' could be misleading. Define the specific types studied and focus on relevant literature to ensure accuracy and clarity.

Clarify the necessity of studying predisposing factors and turnover intention in your setting. Providing this rationale strengthens the significance and relevance of your study.

Please specify the definition of healthcare professionals used in your study for clarity.

Given that you have access to statistics regarding the number of healthcare professionals, it would be appropriate to apply the Cochrane formula with a finite population correction. This adjustment accounts for the specific population size being studied, ensuring more accurate sample size calculations. Please consider referencing the appropriate formula and methodology to support your sample size determination.

Specify the key stakeholders involved in the review process for clarity. Additionally, clarify whether the participants of the pre-test were included or excluded from the subsequent analysis to avoid ambiguity.

Caution is advised when admonishing participants to complete questionnaires quickly, as rushed responses may lead to inaccuracies or misinformation. Providing adequate time for thoughtful consideration can enhance the quality and reliability of the data collected.

Could you please provide information on the reliability and validity checking procedures conducted for the study? Reporting these measures would enhance the credibility and robustness of your findings.

Consider streamlining the methodology section by avoiding repetitive information and focusing on essential details. Prioritize concise descriptions that convey the key aspects of the study design, procedures, and analyses, ensuring clarity without unnecessary repetition.

It appears that some percentages in Table 1 are missing the '%' sign. Please double-check the reported percentages against your output to ensure accuracy and consistency throughout the table.

Could you please provide the rationale behind the classification of variables such as age, highest education level, working experience, type of health facility, type of employment, current position, daily hours of sleep, and working days in a typical week? .

Could you please clarify how you measured awkward posture or good posture in your study? Additionally, for the reported data on Occupational related factors, did you assess the experience over a certain period or the entire service life of the participants? .

In the discussion section, it's advisable to provide an overview of your findings at the beginning. This summary helps orient readers and provides context for the subsequent detailed analysis and interpretation of your results.

Consider providing a more logical explanation in your discussion section by incorporating region-specific studies to support or explain your findings. Focusing on research conducted in the same geographic area can enhance the relevance and applicability of your study's conclusions.

6. PLOS authors have the option to publish the peer review history of their article (what does this mean?). If published, this will include your full peer review and any attached files.

Reviewer #1: No

Reviewer #2: No

---

## [Author Response · Author response to Decision Letter 0]

26 Apr 2024

A file on Response to Reviewers has been attached as part of supporting files.

---

## [Decision Letter · Decision Letter 1]

6 Nov 2024

PONE-D-23-26548R1Prevalence, predisposing factors, and turnover intention related to low back pain among health workers in Accra, GhanaPLOS ONE

Dear Dr. Tawiah,

Thank you for submitting your manuscript to PLOS ONE. After careful consideration, we feel that it has merit but does not fully meet PLOS ONE’s publication criteria as it currently stands. Therefore, we invite you to submit a revised version of the manuscript that addresses the points raised during the review process.

We look forward to receiving your revised manuscript.

Kind regards,

Haruna Musa Moda

Academic Editor

PLOS ONE

Journal Requirements:

Reviewers' comments:

Reviewer's Responses to Questions

**Comments to the Author**

1. If the authors have adequately addressed your comments raised in a previous round of review and you feel that this manuscript is now acceptable for publication, you may indicate that here to bypass the “Comments to the Author” section, enter your conflict of interest statement in the “Confidential to Editor” section, and submit your "Accept" recommendation.

Reviewer #2: All comments have been addressed

Reviewer #3: (No Response)

2. Is the manuscript technically sound, and do the data support the conclusions?

Reviewer #2: Yes

Reviewer #3: Yes

3. Has the statistical analysis been performed appropriately and rigorously? 

Reviewer #2: Yes

Reviewer #3: Yes

4. Have the authors made all data underlying the findings in their manuscript fully available?

Reviewer #2: No

Reviewer #3: Yes

5. Is the manuscript presented in an intelligible fashion and written in standard English?

Reviewer #2: Yes

Reviewer #3: Yes

6. Review Comments to the Author

Reviewer #2: (No Response)

Reviewer #3: My comments: This is a very good study that addresses a critical health issue among healthcare workers in a Ghana. The inclusion of multiple public and private hospitals significantly strengthens the study's findings and relevance.

Specific minor comments in edited version:

• Line 78: The term "neglected" is not accurate here; please consider using a different word.

• Lines 81–83: Please revise the English for clarity.

• Line 92: The word “however” is not the appropriate connector here.

• Line 93: Consider using "the main activities" instead of "some of the…"

• Lines 96–97: The sentence could be improved for clarity. Consider stating the workload intensity of these tasks first, and then, in a separate sentence, note that it is even more exhausting in Africa and other low- and middle-income countries where working aids are lacking (13, 14).

• Line 125: Use "rates" instead of "rate."

• Line 361: Replace "and" with "while" to improve flow.

• Lines 349 and 363: The section title and table title are somewhat vague. Consider editing them to "Occupational Factors Contributing to Lower Back Pain."

• Lines 356 and 373: Clarify these titles as well; I suggest "Organizational, Behavioral, and Intervention-Related Factors Contributing to Lower Back Pain."

Specific questions for edited version:

• Line 245 - Study Questionnaire and Data Collection: How were validity and reliability tested?

• Language of questionnaire: In which language was the questionnaire administered? If it was originally in English, was it translated into the local language of Ghana?

• Table 1: Could you clarify why a majority (84%) of participants are from public hospitals? Is there a particular reason for this distribution?

Major comments for discussion section in edited version:

1. The discussion can be strengthened by explaining more and providing more supporting arguments on why ‘overtime work, on-call duties, understaffing, and working more than five days per week—are significantly associated with increased lower back pain among healthcare workers’

2. Another point that should be further discussed and explained is the “Training on Proper Posture vs. Application”. In table 6, Trained on good working posture was 0.446. This may indicate a gap between training and practical application. The discussion should explore potential reasons, such as high workload, time pressure, or lack of ergonomic support, which might prevent staff from consistently applying proper posture techniques. Furthermore, the authors could consider whether reinforcement mechanisms or regular

3. Unexpected Finding: The finding that heavy lifting is not significantly associated with lower back pain is unexpected, especially given that posture training did not correlate with reduced pain. The authors should consider discussing possible explanations, such as the type or frequency of lifting tasks, use of assistive devices, … etc.

4. The authors should consider performing a statistical analysis or categorization based on the specific domain or role of healthcare staff (e.g., nurses, midwife…) to examine if lower back pain is more prevalent in certain occupational groups. This could reveal whether specific job functions contribute differently to the development of lower back pain.

Major comments for recommendations:

1. Elaborate more on the recommendations at the end of discussion section and conclusion.

o Enhanced Ergonomic Training and Application:

o Regular reminders, visual cues, or periodic "posture refreshers" could be used to help healthcare staff remember to implement proper techniques during work.

o Ensure access to assistive lifting devices, such as mechanical lifts

o Encourage healthcare institutions to foster a culture that prioritizes employee health and safety by regularly addressing occupational health risks.

o Establish a reporting system where staff can communicate ergonomic concerns or report instances of musculoskeletal pain.

7. PLOS authors have the option to publish the peer review history of their article (what does this mean?). If published, this will include your full peer review and any attached files.

Reviewer #2: No

Reviewer #3: **Yes: **Norr Hassan

---

## [Author Response · Author response to Decision Letter 1]

16 Dec 2024

I have attached the "respond to reviewers" document.

---

## [Decision Letter · Decision Letter 2]

2 Jan 2025

Prevalence, predisposing factors, and turnover intention related to low back pain among health workers in Accra, Ghana

PONE-D-23-26548R2

Dear Dr. Tawiah

We’re pleased to inform you that your manuscript has been judged scientifically suitable for publication and will be formally accepted for publication once it meets all outstanding technical requirements.

Kind regards,

Haruna Musa Moda

Academic Editor

PLOS ONE

Additional Editor Comments (optional):

Dear Authors

Many thanks for addressing all suggested improvement by the reviewers.

I am happy to recommend your manuscript be accepted in the present form.

Best wishes

Reviewers' comments:

Reviewer's Responses to Questions

**Comments to the Author**

1. If the authors have adequately addressed your comments raised in a previous round of review and you feel that this manuscript is now acceptable for publication, you may indicate that here to bypass the “Comments to the Author” section, enter your conflict of interest statement in the “Confidential to Editor” section, and submit your "Accept" recommendation.

Reviewer #3: All comments have been addressed

2. Is the manuscript technically sound, and do the data support the conclusions?

Reviewer #3: Yes

3. Has the statistical analysis been performed appropriately and rigorously? 

Reviewer #3: I Don't Know

4. Have the authors made all data underlying the findings in their manuscript fully available?

Reviewer #3: Yes

5. Is the manuscript presented in an intelligible fashion and written in standard English?

Reviewer #3: Yes

6. Review Comments to the Author

Reviewer #3: (No Response)

7. PLOS authors have the option to publish the peer review history of their article (what does this mean?). If published, this will include your full peer review and any attached files.

Reviewer #3: **Yes: **Norr Hassan

---

## [Editor Report · Acceptance letter]

7 Jan 2025

PONE-D-23-26548R2 

PLOS ONE

Dear Dr. Tawiah, 

I'm pleased to inform you that your manuscript has been deemed suitable for publication in PLOS ONE. Congratulations! Your manuscript is now being handed over to our production team.

Kind regards, 

on behalf of

Dr. Haruna Musa Moda 

Academic Editor

PLOS ONE